# SupertonicTTS: Towards Highly Efficient and Streamlined Text-to-Speech System

## Abstract

We introduce SupertonicTTS, a novel text-to-speech (TTS) system designed for efficient and streamlined speech synthesis. SupertonicTTS comprises three components: a speech autoencoder for continuous latent representation, a text-to-latent module leveraging flow-matching for text-to-latent mapping, and an utterance-level duration predictor. To enable a lightweight architecture, we employ a low-dimensional latent space, temporal compression of latents, and ConvNeXt blocks. The TTS pipeline is further simplified by operating directly on raw character-level text and employing cross-attention for text-speech alignment, thus eliminating the need for grapheme-to-phoneme (G2P) modules and external aligners. In addition, we propose context-sharing batch expansion that accelerates loss convergence and stabilizes text-speech alignment with minimal memory and I/O overhead. Experimental results demonstrate that SupertonicTTS delivers performance comparable to contemporary zero-shot TTS models with only 44M parameters, while significantly reducing architectural complexity and computational cost.

## 1 Introduction

Text-to-speech (TTS) technology has made remarkable advancements in recent years, unlocking groundbreaking capabilities and enhancing user experiences. For instance, modern TTS models can synthesize natural voice for speakers unseen during training (Casanova et al., 2022; Saeki et al., 2023). This is achieved with only a few adaptation steps using a small amount of data (Huang et al., 2022; Kim et al., 2022) or even without any fine-tuning at all, a feature referred to as zero-shot capability (Jiang et al., 2024; Kim et al., 2024a). Moreover, modern TTS systems provide a wide range of powerful functionalities within a single model, such as voice conversion (Kim et al., 2021), multilingual synthesis (Casanova et al., 2024), content editing (Tae et al., 2021; Peng et al., 2024), and noise removal (Le et al., 2024).

Despite significant technical achievements, however, most contemporary TTS systems still rely on a massive number of parameters and complex pipelines that include a grapheme-to-phoneme (G2P) module, a text-speech aligner, or pretrained models for extracting textual and speaker features, as outlined in Table 1. These factors collectively contribute to increased computational overhead during both training and inference, and introduce complex interdependencies among system components. Given these challenges, a promising research direction in TTS is the development of **a more streamlined pipeline that reduces architectural complexity and computational cost while maintaining competitive performance.**

To this end, we propose **SupertonicTTS**, a novel TTS system designed to deliver high-quality speech with exceptional efficiency and a streamlined process. This system consists of three modules: (1) **a speech autoencoder** that encodes audio into a continuous latent representation, (2) **a text-to-latent module** that maps text and speaker information to corresponding latents using a flow-matching algorithm (Lipman et al., 2023), and (3) **a duration predictor** that estimates the total duration of speech to be synthesized. This work focuses on the careful design of these modules to achieve a highly efficient and simplified TTS system.

We also introduce several techniques to enhance architectural flexibility, improve training stability, and reduce model complexity. First, we design the latent space with a remarkably low dimensionality and compress the latents along the temporal axis before passing them to the text-to-latent module. This strategy enables the decoupling of high-resolution speech synthesis from low-resolution latent

Table 1: Comparison of SupertonicTTS with contemporary text-to-speech models ("PD": phoneme-level duration requirement, "TSA": use of a text-to-speech aligner during training, "AR": autoregressive inference over text input, "LR": use of a length regulator during inference, "RT": transcription requirement for reference speech during inference, "TP": text preprocessor, "SR": sampling rate, "#Param.": total parameter count including duration predictor and vocoder). † indicates that the number is an estimate based on architecture descriptions in the baseline papers.

| | PD | TSA | AR | LR | RT | TP | SR (Hz) | #Param. |
|---|---|---|---|---|---|---|---|---|
| Wang et al. (2023) | ✗ | ✗ | ✓ | ✗ | ✓ | G2P | 24,000 | 410M† |
| Le et al. (2024) | ✓ | ✓ | ✗ | ✓ | ✓ | G2P | 16,000 | 371M |
| Jiang et al. (2024) | ✓ | ✓ | ✓ | ✓ | ✗ | G2P | 16,000 | 473M |
| Kim et al. (2024a) | ✗ | ✗ | ✓ | ✗ | ✓ | ByT5 (Xue et al., 2022) | 22,050 | >1.3B† |
| Lee et al. (2025) | ✗ | ✗ | ✗ | ✗ | ✓ | SpeechT5 (Ao et al., 2022) | 22,050 | 970M |
| Ours | ✗ | ✗ | ✗ | ✗ | ✗ | **Raw** | **44,100** | **44M** |

modeling. Second, we introduce context-sharing batch expansion to achieve the benefits of a larger batch size with minimal computational overhead. Third, we employ ConvNeXt blocks (Liu et al., 2022; Siuzdak, 2024; Okamoto et al., 2023) extensively across all modules to ensure a lightweight and efficient architecture. In addition to these primary contributions, we simplify the TTS pipeline by employing cross-attention mechanisms for text-speech alignment, and by using raw characters as input. Furthermore, we refrain from incorporating external pretrained models, thereby reducing architectural dependencies and complexity.

Through extensive experiments, we rigorously evaluate SupertonicTTS, confirming its competitive performance combined with simplicity and efficiency. First, we demonstrate that speech can be encoded into a low-dimensional latent space and reconstructed with high fidelity at remarkable speed using a ConvNeXt-based architecture. Second, we show that context-sharing batch expansion enhances both loss convergence and alignment learning in the text-to-latent module, even surpassing batch size scaling. Finally, we show that SupertonicTTS achieves competitive zero-shot TTS performance with just 44 million parameters and extremely fast generation.

## 2 RELATED WORK

Modern TTS systems have been developed through various approaches. One major direction utilizes signal processing features, such as mel spectrograms, as an intermediate representation (Jeong et al., 2021; Kim et al., 2020; Kong et al., 2020; Lee et al., 2023; Le et al., 2024). This approach allows for modular system design where an acoustic model converts text into these features and a vocoder synthesizes a waveform from them. While this simplifies implementation, reliance on hand-crafted features limits the exploitation of latent space and constrains the model's representational capacity. Another common approach uses discrete tokens from neural audio codec models (Kharitonov et al., 2023; Kim et al., 2024a; Wang et al., 2023). By leveraging language modeling techniques, this method improves the naturalness, intelligibility, and speaker similarity of synthesized speech. However, errors from the vector-quantization step can degrade speech quality, which is especially critical at low bit-rates. Additionally, the use of residual vector quantization (RVQ) (Défossez et al., 2023; Zeghidour et al., 2022) often adds architectural complexity by requiring a prediction of multiple tokens per frame. A third approach focuses on disentangled latent spaces for fine-grained control over diverse attributes of the generated speech (Ju et al., 2024; Choi et al., 2023; Polyak et al., 2021). In this method, speech is encoded into distinct features such as linguistic content, speaker identity, and prosody, and TTS models are trained to estimate these disentangled features. However, achieving effective disentanglement often requires intricate loss objectives, integrating pretrained models, and a large number of parameters. These requirements can increase engineering efforts and model latency.

In pursuit of efficient and simplified TTS, recent works have sought to avoid complex components such as G2P modules, phoneme-level duration modeling, and explicit text-speech aligners (Lovelace et al., 2024; Eskimez et al., 2024; Yang et al., 2024a; Lee et al., 2025; Chen et al., 2024). Nevertheless, they still rely on text encoders pretrained for specific languages or suffer from training instabil-

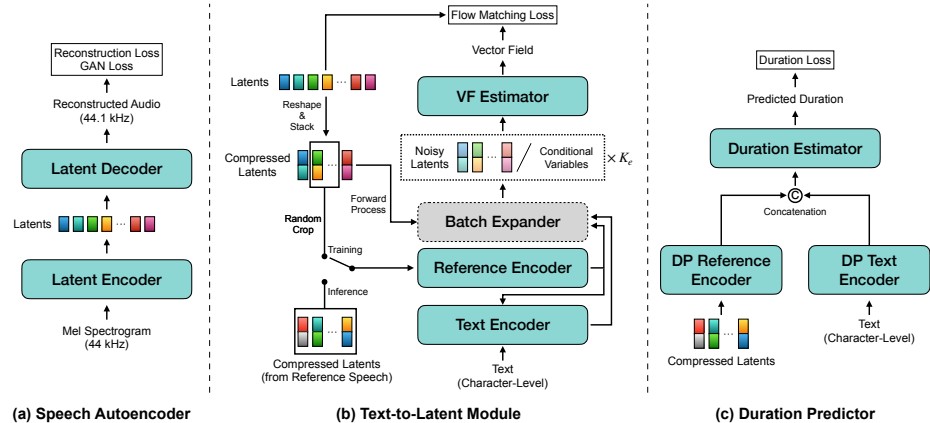

Figure 1: Overall architecture of SupertonicTTS.

ity due to the difficulty of learning text-speech alignment. In contrast, our approach overcomes these limitations by operating directly on character-level input without relying on pretrained text encoders and by introducing context-sharing batch expansion to accelerate and stabilize alignment learning.

## 3 METHOD

SupertonicTTS is built on latent diffusion models (LDMs), which have shown state-of-the-art performance in different generative tasks (Rombach et al., 2022; Lovelace et al., 2023; Podell et al., 2024). More specifically, the training of SupertonicTTS is divided into three phases: first, a speech autoencoder is trained to map input audio into a low-dimensional latent space and reconstruct the original audio. Next, a text-to-latent module learns to generate latent representations that accurately reflect the speech characteristics of both input text and reference speech. Finally, a duration predictor is optimized to estimate the total speech duration based on input text and reference speech. The overall architecture of SupertonicTTS is depicted in Fig. 1.

### 3.1 SPEECH AUTOENCODER

The speech autoencoder converts input audio into latent representations using a latent encoder and reconstructs the audio from these representations with a latent decoder. In this work, we use mel spectrograms as input features to the latent encoder, rather than raw audio. Our preliminary experiments show that this approach accelerates the convergence of the training loss. The latent space is designed to be continuous and **significantly lower in dimensionality** than the number of mel spectrogram channels. The input-output structure of the speech autoencoder aligns with that of conventional neural vocoders (Kong et al., 2020; Lee et al., 2023). Therefore, the speech autoencoder can be interpreted as a neural vocoder with a compact latent space in its middle.

### 3.1.1 ARCHITECTURE

The latent encoder is built upon the Vocos architecture, which is primarily composed of ConvNeXt blocks for improved computational efficiency (Siuzdak, 2024; Liu et al., 2022). To tailor the architecture for latent encoding, we remove the original Fourier head and introduce a linear layer just before the final normalization. This linear layer projects hidden representations into a lower-dimensional space. Although the latent encoder is not used during TTS inference, its efficient design is leveraged to enable fast latent encoding throughout the training of the text-to-latent module.

Similarly, the latent decoder follows the Vocos architecture with several key modifications. First, we adapt a depth-wise convolution layer in the ConvNeXt blocks to be causal and dilated. The introduction of causal layers allows the latent decoder to operate in streaming mode. Additionally, instead of using the Fourier head, we introduce two linear layers with PReLU activation (He et al., 2015). The final time-domain output is derived by flattening the frame-level output. This approach

is inspired by WaveNeXt (Okamoto et al., 2023), but we adopt a higher hidden dimensionality and nonlinearity to enhance representational capacity.

Architectural details for the speech autoencoder are provided in Appendix A.1.

### 3.1.2 OPTIMIZATION

Similar to modern neural vocoders (Lee et al., 2023; Kong et al., 2020), the speech autoencoder is trained within a Generative Adversarial Network (GAN) framework (Goodfellow et al., 2014), using a combination of reconstruction loss $\mathcal{L}_{\text{recon}}$, adversarial loss $\mathcal{L}_{\text{adv}}$, and feature matching loss $\mathcal{L}_{\text{fm}}$. The reconstruction loss is defined as the multi-resolution spectral $L_1$ loss, computed in the mel spectrogram domain. For adversarial training, both multi-period discriminators (MPD) (Kong et al., 2020) and multi-resolution discriminators (MRD) (Jang et al., 2021) are employed to enhance perceptual quality. Additionally, the feature matching loss is applied to minimize the $L_1$ distances between the discriminator features of real and generated samples, thereby further stabilizing the adversarial training process. The final loss for the training phase of generator is given by $\mathcal{L}_{\text{G}} = \lambda_{\text{recon}}\mathcal{L}_{\text{recon}} + \lambda_{\text{adv}}\mathcal{L}_{\text{adv}} + \lambda_{\text{fm}}\mathcal{L}_{\text{fm}}$. Further details on the discriminator architecture and objective functions can be found in Appendices A.1.3 and B.1.

## 3.2 TEXT-TO-LATENT MODULE

The text-to-latent module generates a latent representation that captures essential speech characteristics from both input text and reference speech, following the LDM framework (Lovelace et al., 2023; Podell et al., 2024; Rombach et al., 2022). Specifically, an initial noise $z_0$ is drawn from a base distribution $p(z_0)$, typically set as a simple prior such as $\mathcal{N}(0, I)$. The noise is then progressively refined into a structured representation $z_t$ through a time-dependent vector field induced by the flow-matching framework (Lipman et al., 2023). The text-to-latent module estimates this vector field based on text, reference speech, and time $t$, ensuring that the final representation $z_1$ preserves the relevant speech attributes.

Unlike most recent state-of-the-art TTS models, the text-to-latent module **does not rely on external pretrained models, G2P modules, or text-to-speech aligners**. Specifically, it uses character-level text as input and employs cross-attention mechanisms to align text and speech within a streamlined architecture. To improve architectural flexibility and training stability, we also introduce two novel techniques: **temporal compression of latents** and **context-sharing batch expansion**.

### 3.2.1 TEMPORALLY COMPRESSED LATENT REPRESENTATION

We propose to decouple high-resolution audio synthesis from lower-rate latent modeling via temporal compression. Specifically, given a compression factor $K_c$, we transform a latent of shape $(C, T)$ into a tensor of shape $(K_c C, \frac{T}{K_c})$, where $C$ is the original latent dimensionality and $T$ denotes the number of temporal frames. In our implementation, we set $K_c = 6$ to align the speech autoencoder with a frame rate of around $86\,\text{Hz}$, which is consistent with typical vocoder settings (Lee et al., 2023; Choi et al., 2023), and the text-to-latent module with a lower rate of roughly $14\,\text{Hz}$, following common settings in semantic token models (Kim et al., 2024a; Kharitonov et al., 2023). We also set $C = 24$, resulting in a descent channel size of $K_c C = 144$ for the text-to-latent module.

This approach provides several advantages over using the original latents. First, it reduces computational costs, which is particularly advantageous for layers that rely on computationally intensive sequential operations such as the attention mechanism (Vaswani et al., 2017). Second, it alleviates the text-speech alignment challenge by reducing the total number of speech frames. Finally, all temporal information is preserved in the transformed representation, allowing for perfect inversion.

### 3.2.2 CONTEXT-SHARING BATCH EXPANSION

We propose context-sharing batch expansion to improve the training efficiency of conditional generative models based on diffusion or flow-matching algorithms (Ho et al., 2020; Lipman et al., 2023). In conventional training, an input variable is perturbed with random noise at a sampled timestep (forward process), and the model is optimized to denoise it using the corresponding conditioning variables (reverse process). In contrast, given an expansion factor $K_e$, the proposed method generates

$K_e$ perturbed inputs by sampling $K_e$ noise-timestep pairs, while reusing the same conditions across all $K_e$ samples. This strategy mimics the effect of increasing the batch size but is computationally more efficient when conditions are pre-encoded. Importantly, we empirically demonstrate that our method improves the learning of text-speech alignment more effectively than simply increasing the batch size. A pseudo-algorithm is provided in Appendix C for further clarification.

### 3.2.3 ARCHITECTURE

The reference encoder takes latent representations from a reference speaker as input. These latents are obtained by cropping a portion of input speech during training, and extracted from reference speech during inference. It then processes the latents using multiple ConvNeXt blocks and generates reference key and value vectors through two attention layers, following the timbre token block introduced in NANSY++ (Choi et al., 2023). Note that the reference key and value are fixed-size vectors, independent of the input length.

The text encoder processes character-level input using ConvNeXt blocks and self-attention layers. This architecture is designed to efficiently capture both local and long-range dependencies in text, offering computational efficiency. The output from the self-attention layers is further refined using reference key and value vectors through two cross-attention layers, producing speaker-adaptive text representations. A key design choice is the exclusion of G2P and other pretrained modules, ensuring the model learns everything directly from the character input.

The vector field (VF) estimator is primarily composed of ConvNeXt blocks, along with time-conditioning, text-conditioning, and reference-conditioning blocks. To enhance model expressiveness, certain ConvNeXt blocks incorporate dilated convolutional layers. Time conditioning is applied by globally adding a time embedding to the input sequence. Both text and reference conditioning utilize cross-attention, where the conditional variables serve as keys and values.

More details on the architecture of the text-to-latent module can be found in Appendix A.2.

### 3.2.4 OPTIMIZATION

The text-to-latent module is optimized using the flow-matching algorithm (Lipman et al., 2023). During training, a randomly cropped segment of the compressed latents serves as input to the reference encoder. To prevent information leakage from this, we apply a mask to the corresponding segment when calculating the flow-matching loss, similar to previous work (Lee et al., 2025; Kim et al., 2024b). Specifically, our optimization objective is as follows:

$$\mathcal{L}_{\text{TTL}} = \mathbb{E}_{t,(z_1,c),p(z_0)} \| \boldsymbol{m} \cdot (v(z_t, z_{\text{ref}}, c, t) - (z_1 - (1 - \sigma_{\min})z_0)) \|_1, \tag{1}$$

where $v$, $\boldsymbol{m}$, $z_1$, $z_0$, $z_t$, $z_{\text{ref}}$, and $c$ represent the text-to-latent module, the reference mask, compressed latents, noise sampled from the base distribution $p(z_0)$, noisy latents $z_t = (1 - (1 - \sigma_{\min})t)z_0 + tz_1$, cropped latents $z_{\text{ref}} = (1 - \boldsymbol{m}) \cdot z_1$, and text, respectively. We set $t \sim \mathcal{U}[0, 1]$ and $p(z_0) = \mathcal{N}(0, 1)$. Furthermore, with a probability of $p_{\text{uncond}}$, the model is trained without conditions $z_{\text{ref}}$ and $c$ to enable classifier-free guidance (CFG) (Ho & Salimans, 2021). In this unconditional mode, the conditioning variables are replaced with learnable parameters.

### 3.3 DURATION PREDICTOR

At inference time, the proposed framework requires **the total length of latent representations** to be synthesized. In this context, SupertonicTTS is expected to be robust to errors in duration estimation, compared to other TTS models that rely on phoneme-level durations (Kim et al., 2021; 2024b; Le et al., 2024; Yang et al., 2024b). With this in mind, we design **an utterance-level duration predictor with a simple, lightweight architecture**. Specifically, an utterance-level text embedding and a reference embedding are obtained using a small number of ConvNext blocks and attention layers. These embeddings are concatenated and transformed into a scalar value representing the total speech duration via linear layers, resulting in a total parameter count of approximately 0.5M. The duration predictor is trained using the $L_1$ distance between the ground truth and predicted durations. Further details are provided in Appendix A.3.

Table 2: Evaluation of reconstruction quality and inference speed on *LT-clean* and *LT-other*.

| | LT-clean | | | LT-other | | | |
|---|---|---|---|---|---|---|---|
| | NISQA | UTMOSv2 | V/UV F1 | NISQA | UTMOSv2 | V/UV F1 | RTF |
| GT | 4.09 ± 0.03 | **3.26 ± 0.02** | - | 3.61 ± 0.03 | **3.01 ± 0.02** | - | - |
| BigVGAN | **4.11 ± 0.03** | 3.16 ± 0.02 | **0.9735** | 3.61 ± 0.03 | 2.85 ± 0.02 | **0.9620** | 0.0124 (RTX 4090) |
| Ours | 4.06 ± 0.03 | 3.13 ± 0.02 | 0.9587 | **3.76 ± 0.03** | 2.88 ± 0.02 | 0.9450 | **0.0006** (RTX 4090) |

## 4 TRAINING SETUP

### 4.1 DATASET

We trained the speech autoencoder with a combined collection of publicly available datasets and our internal database, resulting in a total of 11,167 hours of audio recordings from approximately 14,000 speakers. A detailed list of the public datasets is provided in Appendix F. For the training of the text-to-latent module and the duration predictor, we selected four English datasets: LJSpeech (Ito & Johnson, 2017), VCTK (Yamagishi et al., 2019), Hi-Fi TTS (Bakhturina et al., 2021), and LibriTTS (Zen et al., 2019). Collectively, these datasets encompass 945 hours of high-quality speech from 2,576 English speakers. All audio files were resampled to a target sample rate of $44.1\,\mathrm{kHz}$, if their original sample rate differed.

### 4.2 OPTIMIZATION

We optimized the speech autoencoder for 1.5M iterations using the AdamW optimizer (Loshchilov & Hutter, 2019) with a learning rate of $2 \times 10^{-4}$ and a batch size of 128. We used four NVIDIA RTX 4090 GPUs. The loss function coefficients were configured as follows: $\lambda_{\mathrm{recon}} = 45$, $\lambda_{\mathrm{adv}} = 1$, and $\lambda_{\mathrm{fm}} = 0.1$. For adversarial training, we randomly cropped segments of real and generated speech to $0.19\,\mathrm{s}$. The log-scaled mel spectrogram input was obtained using an FFT size of 2048 ($46.43\,\mathrm{ms}$), a Hann window of the same size, a hop size of 512 ($11.61\,\mathrm{ms}$), and 228 mel bands. Additional details on the optimization of the speech autoencoder are provided in Appendix B.1.

The text-to-latent module was optimized using the AdamW optimizer for 700k iterations with a batch size of 64 and an expansion factor $K_e = 4$. The learning rate was initially set to $5 \times 10^{-4}$ and halved every 300k iterations. Training was conducted on four RTX 4090 GPUs. Latents were normalized using precomputed channel-wise mean and variance statistics before being input into the text-to-latent module. We set $p_{\mathrm{uncond}} = 0.05$ and $\sigma_{\min} = 10^{-8}$. During training, reference speech segments were obtained by randomly cropping the input audio, with durations ranging from $0.2\,\mathrm{s}$ to $9\,\mathrm{s}$. We ensured that the cropped length did not exceed half of the original speech duration.

The duration predictor was trained for 3,000 iterations using the AdamW optimizer with a learning rate of $5 \times 10^{-4}$ and a batch size of 128 on a single RTX 4090 GPU. During training, reference speech was obtained by randomly selecting a segment from 5% to 95% of the input speech.

## 5 EXPERIMENTS

We used four test sets for evaluation, each containing audio samples ranging from 4 to 10 seconds: *LT-clean*, *LT-other*, *LS-clean*, and *LS-PC-clean*. *LT-clean* and *LT-other* were derived from the test-clean and test-other sets of LibriTTS (Zen et al., 2019), respectively. *LS-clean* was sourced from the test-clean set of LibriSpeech (Panayotov et al., 2015), while *LS-PC-clean* corresponds to the test set proposed by Chen et al. (2024). For the text-to-latent module, we set the number of function evaluations (NFE) to 32, with its effect analyzed in Appendix D.1.

### 5.1 SPEECH RECONSTRUCTION

We evaluated reconstruction quality by comparing the speech autoencoder with the official $44.1\,\mathrm{kHz}$ checkpoint of BigVGAN (Lee et al., 2023). Experiments were conducted on *LT-clean* and *LT-other* using three metrics: NISQA (Mittag et al., 2021), UTMOSv2 (Baba et al., 2024), and CREPE (Kim et al., 2018). Both NISQA and UTMOSv2 are mean opinion score (MOS) prediction systems to

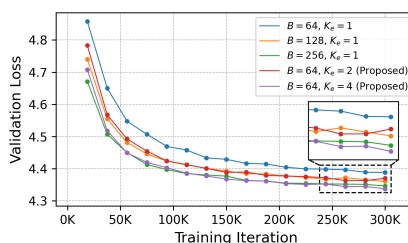 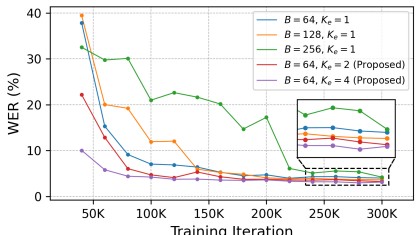

Figure 2: Comparison of the proposed batch expansion and increase in batch size.

estimate the perceptual quality of speech signals. CREPE is used to analyze pitch in speech and compute the F1 score for voiced/unvoiced classification (V/UV F1), which serves as a key metric for identifying artifacts in generated speech. Additionally, we measured the real-time factor (RTF) on an RTX 4090 GPU to quantify inference speed.

We report the results in Table 2. In terms of NISQA scores, the speech autoencoder performs competitively, achieving 4.06 on *LT-clean* and outperforming BigVGAN on *LT-other* with a score of 3.76. Similarly, in terms of UTMOSv2 scores, the speech autoencoder slightly lags behind BigV-GAN *LT-clean* but surpasses it on *LT-other*. These results support that the speech autoencoder can synthesize perceptually high-quality speech. For the F1 evaluation of V/UV classification, BigV-GAN achieves the highest scores (0.9735 on *LT-clean* and 0.9620 on *LT-other*), while the speech autoencoder follows closely with 0.9587 and 0.9450, respectively. This suggests that our model introduces minor artifacts affecting voiced/unvoiced classification. Nonetheless, the overall performance remains competitive considering the V/UV F1 scores of other neural vocoders reported in the baseline paper.[1] Notably, the speech autoencoder achieves an inference speed more than 20 times faster than BigVGAN. These results collectively confirm that our speech autoencoder, despite its bottlenecked architecture and low-dimensional latent space, can generate high-quality speech while offering a substantial advantage in inference efficiency.

## 5.2 EVALUATION OF CONTEXT-SHARING BATCH EXPANSION

To assess the effectiveness of context-sharing batch expansion, we trained four additional models using the following combinations of batch size $B$ and expansion factor $K_e$: (64, 1), (128, 1), (256, 1), and (64, 2). Performance was evaluated with validation loss and pronunciation error. Validation loss was computed on *LT-clean* by averaging $\mathcal{L}_{\text{TTL}}$ across five timesteps: 0.1, 0.3, 0.5, 0.7, and 0.9. Pronunciation error was quantified using word error rate (WER) and character error rate (CER) between synthesized speech transcriptions and the ground-truth. For this evaluation,

Table 3: Final error rates on *LS-clean*.

| $B$ | $K_e$ | WER(%) | CER(%) |
|---|---|---|---|
| 64 | 1 | 3.11 | 1.08 |
| 128 | 1 | 3.08 | 1.06 |
| 64 | 2 | **2.97** | **1.00** |
| 256 | 1 | 2.88 | 0.92 |
| 64 | 4 | **2.64** | **0.83** |

we generated five samples per utterance from *LS-clean* and transcribed them using a CTC-based HuBERT-Large model (Hsu et al., 2021). All transcriptions were normalized using NVIDIA's NeMo-text-processing (Zhang et al., 2021) before computing WER and CER.

Fig. 2 shows validation loss and WER curves throughout training up to 300k iterations. It can be observed that the validation loss converges faster as $K_e$ increases, similar to the effect of increasing $B$. Interestingly, WER improves more rapidly with higher $K_e$, whereas increasing $B$ actually degrades performance, particularly in the early stages of training (e.g., around 100k iterations). Although this gap narrows as training progresses, models trained with larger $K_e$ ultimately

Table 4: Computational efficiency comparison.

| $B$ | $K_e$ | Memory | Iter. Time | GFLOPs |
|---|---|---|---|---|
| 16 | 1 | 2.47 | 0.083s | 65.295 |
| 32 | 1 | 4.53 | 0.149s | 130.59 |
| 16 | 2 | **3.96** | **0.098s** | **120.07** |
| 64 | 1 | 8.61 | 0.293s | 261.18 |
| 16 | 4 | **6.92** | **0.136s** | **229.63** |

achieve lower final WER and CER than those with correspondingly larger $B$, as summarized in Ta-

---

[1]Lee et al. (2023) state that HiFi-GAN (Kong et al., 2020) and WaveFlow (Ping et al., 2020) achieve V/UV F1 scores of 0.9300 and 0.9410, respectively, on the dev subsets of LibriTTS.

Table 5: Performance comparison with contemporary zero-shot TTS systems. "Data" refers to the total amount of transcribed speech (in hours) used for training, with entries marked (\*) denoting a multilingual dataset. "#DP", "#T2F", "#F2S" and "#All" represent the number of parameters in the duration predictor, text-to-feature module (e.g., text-to-mel, text-to-latent), feature-to-speech module (e.g., vocoder), and the entire text-to-speech model, respectively. Parameter counts marked with a dagger ($\dagger$) are estimated from the architectural descriptions in the baseline papers. RTF is measured on 10-second audio synthesis.

| Test set | Model | WER | CER | Data | #DP | #T2F | #F2S | #All | RTF |
|---|---|---|---|---|---|---|---|---|---|
| | GT | 2.18 | 0.60 | - | - | - | - | - | - |
| *LS-clean* | VALL-E | 5.9 | - | 60k | - | 403M$^\dagger$ | 7M | 410M$^\dagger$ | $\sim$0.64 |
| | VoiceBox | **1.9** | - | 60k | 28M | 330M | 13M | 371M$^\dagger$ | $\sim$0.62 |
| | CLaM-TTS | 5.11 | 2.87 | 55k | - | >1.23B$^\dagger$ | 112M | >1.3B$^\dagger$ | 0.42 (A100) |
| | DiTTo-TTS | 2.56 | 0.89 | 55k | 33M | 825M | 112M | 940M | 0.16 (A100) |
| | Ours | 2.64 | **0.83** | 945 | 0.5M | 18.5M | 25M | **44M** | **0.02** (RTX 4090) |
| | GT | 1.86 | 0.50 | - | - | - | - | - | - |
| *LS-PC-clean* | FireRedTTS | 2.69 | - | 248k\* | - | 538M | 235M | 773M | 0.84 (RTX 3090) |
| | F5-TTS | 2.42 | - | 100k\* | - | 335.8M | 13.5M | 349M | 0.31 (RTX 3090) |
| | Ours | **2.41** | **0.80** | 945 | 0.5M | 18.5M | 25M | **44M** | **0.05** (RTX 3090) |

ble 3. We hypothesize that presenting the same text-speech pair with varying noise and timesteps is more effective for alignment learning compared to using different text–speech pairs. Complete evaluation curves for the entire training process are provided in Appendix D.2.

We also assessed the computational efficiency of context-sharing batch expansion by analyzing GPU memory usage (GiB), time per iteration (seconds), and the number of floating-point operations (GFLOPs). For testing, we used 15-second input speech, 250 characters, and 3-second reference speech. Iteration times were measured over 100 trials using a single RTX 4090 GPU, with 95% confidence intervals narrower than $0.2\,\mathrm{ms}$. GPU memory usage and iteration time were measured for a single training iteration, while GFLOPs were calculated from a single forward pass through the text-to-latent module. Table 4 shows that increasing $K_e$ consistently offers better efficiency compared to increasing $B$ across all metrics. Notably, raising $K_e$ from 1 to 4 results in a 64% increase in iteration time, whereas increasing $B$ by a factor of 4 leads to a 253% increase. The overall results demonstrate that the proposed method not only stabilize text-speech alignment but also significantly enhances training efficiency with respect to memory usage and processing time.

## 5.3 COMPARISON WITH OTHER ZERO-SHOT TTS MODELS

In this section, we provide a comparative analysis of our proposed model against state-of-the-art TTS systems. Specifically, we evaluate our model against six baselines: VALL-E (Wang et al., 2023), VoiceBox (Le et al., 2024), CLaM-TTS (Kim et al., 2024a), DiTTo-TTS (Lee et al., 2025), FireRedTTS (Guo et al., 2024), and F5-TTS (Chen et al., 2024). These models exhibit exceptional TTS performance even in zero-shot scenarios, establishing them as strong benchmarks for our study.

Table 5 presents an overall comparison of pronunciation errors (WER and CER), the amount of transcribed speech used for training (Data), model parameter counts (#DP, #T2F, #F2S, and #All), and inference speed (RTF). Baseline results are sourced from their publications. On the *LS-clean* benchmark, SupertonicTTS achieves a WER of 2.64 and the lowest CER of 0.83, demonstrating strong pronunciation accuracy. The performance is competitive with much larger systems such as DiTTo-TTS (WER 2.56, CER 0.89) and VoiceBox (WER 1.9), despite a much smaller parameter count. Also, our model achieves the lowest WER of 2.41 on *LS-PC-clean*, outperforming FireRedTTS (WER 2.69) and F5-TTS (WER 2.42). While data efficiency is not the primary focus of this study, it is noteworthy that our system achieves such results with a training corpus that is orders of magnitude smaller than those used by other systems. From the perspective of model size, SupertonicTTS is remarkably compact. The entire system contains only 44M parameters, compared to hundreds of millions or even billions in the baselines. In particular, our text-to-latent module (#T2F) requires 18.5M parameters, whereas the next smaller model, VoiceBox, allocates 330M parameters, which is approximately 18 times larger. This parameter efficiency highlights the effectiveness of our architectural choices in capturing linguistic and acoustic mappings with minimal overhead. Another

Table 6: Preference test results and reference speech quality used for zero-shot TTS evaluation.

| | Preference test | | Reference quality | |
|---|---|---|---|---|
| | Naturalness | Similarity | PESQ | SI-SDR |
| Ours vs. VALL-E | **0.233 ± 0.112** | 0.043 ± 0.115 | 3.422 ± 0.327 | 22.526 ± 2.884 |
| Ours vs. VoiceBox | -0.476 ± 0.106 | **0.316 ± 0.113** | 2.171 ± 0.377 | 10.978 ± 8.851 |
| Ours vs. CLaM-TTS | 0.084 ± 0.106 | 0.076 ± 0.113 | 3.642 ± 0.359 | 23.071 ± 3.789 |
| Ours vs. DiTTo-TTS | 0.076 ± 0.109 | 0.057 ± 0.112 | 3.746 ± 0.253 | 24.799 ± 2.034 |

key strength of our system lies in inference speed. With an RTF of 0.02 on an RTX 4090 and 0.05 on an RTX 3090, our model runs substantially faster than all baselines, which report RTFs between 0.16 and 0.84. Such efficiency makes our approach well-suited for real-time or resource-constrained deployment scenarios, where large-scale TTS systems often face limitations.

We also conducted subjective listening tests via Amazon Mechanical Turk to assess perceptual quality. Specifically, we compared SupertonicTTS against VALL-E, VoiceBox, CLaM-TTS, and DiTTo-TTS using audio samples from their respective demo pages. For each comparison, we performed preference tests on 15 paired samples with 42 participants. Each pair consisted of one sample from a baseline model and one from SupertonicTTS. Participants evaluated the pairs based on two criteria: naturalness (which sample sounds more natural) and speaker similarity (which sample sounds more similar to the reference speech). For quantitative analysis, responses were recorded on a five-point scale from -2 to +2, where positive scores indicate a preference for SupertonicTTS. Detailed instructions provided to the participants are included in Appendix E.

Table 6 summarizes the results and shows that SupertonicTTS delivers highly competitive performance. For naturalness, SupertonicTTS was clearly preferred over VALL-E (0.233) and slightly preferred over CLaM-TTS (0.084) and DiTTo-TTS (0.076). However, the subjects perceived Voice-Box samples as more natural than those from SupertonicTTS (-0.476). In terms of speaker similarity, SupertonicTTS was strongly favored over VoiceBox (0.316) and slightly outperformed the other baselines. The contrasting results relative to VoiceBox prompted further investigation. We analyzed the reference speech samples used in the subjective listening test. Specifically, we measured perceptual evaluation of speech quality (PESQ) (Rix et al., 2001) and scale-invariant signal-to-distortion ratio (SI-SDR) of these samples, which is also reported in Table 6. The analysis revealed that VoiceBox's reference audio had significantly lower PESQ (2.171) and SI-SDR (10.978) than the comparable-quality references for the other baselines (PESQ > 3.4 and SI-SDR > 22). Based on these findings, we conjecture that SupertonicTTS reproduces the characteristics of a given reference more faithfully than VoiceBox, but the low reference quality likely caused the SupertonicTTS output to be judged as less natural, despite its strong speaker similarity. Overall, these experimental results support that SupertonicTTS achieves competitive performance on par with recent state-of-the-art zero-shot TTS models. We encourage readers to visit our demo page[2] for a perceptual comparison.

# 6 CONCLUSION

In this paper, we introduced SupertonicTTS, a novel text-to-speech system designed to effectively address the limitations of contemporary TTS models. Within the LDM framework, we designed a system comprising a speech autoencoder, flow-matching-based text-to-latent module, and utterance-level duration predictor. To enhance efficiency and simplicity, we incorporated a low-dimensional latent space, latent compression, context-sharing batch expansion, and ConvNeXt blocks. Furthermore, we simplified the pipeline by eliminating external dependencies such as G2P modules, text-speech aligners, and pretrained text encoders. Our extensive experiments validated that SupertonicTTS provides competitive zero-shot TTS performance with only 44 million parameters and fast inference speed. We believe the proposed system substantially reduces architectural complexity and computational overhead in speech synthesis, opening promising avenues for future research in diverse speech applications and real-time scenarios.

---

[2]https://yfqtylmi.github.io/.

ETHICS STATEMENT

This research involved human participants through a subjective listening test on Amazon Mechanical Turk as detailed in Section 5.3 and Appendix E. An internal peer review confirmed that our human evaluation complies with the ICLR Code of Ethics. Participants were informed about the task through the standard MTurk interface and accompanying instructions.

REPRODUCIBILITY STATEMENT

The paper provides detailed descriptions of the model architecture (Section 3, Appendix A), datasets used (Section 4.1, Appendix F), and optimization procedures (Section 4.2, Appendix B). This information should allow for a high degree of reproducibility for the main experimental results.

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

# A ARCHITECTURE DETAILS

## A.1 SPEECH AUTOENCODER

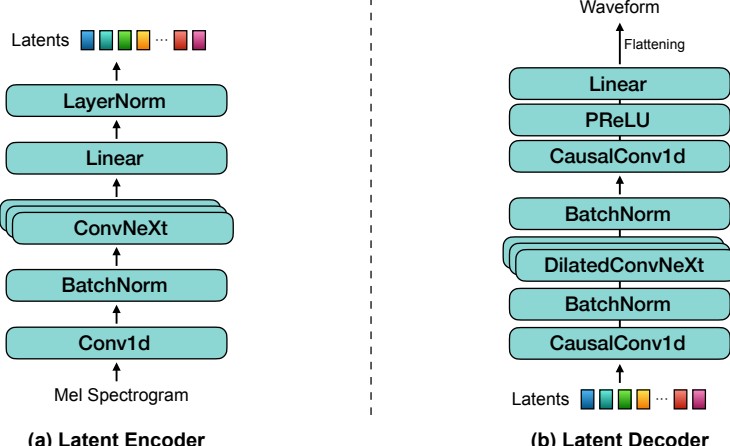

**(a) Latent Encoder**          **(b) Latent Decoder**

Figure 3: Detailed architecture of latent encoder and latent decoder in speech autoencoder.

We illustrate the detailed architecture of the speech autoencoder in Fig. 3. Both the latent encoder and the latent decoder are built on the Vocos architecture (Siuzdak, 2024) with several modifications, aiming for useful applications (e.g., latent encoding, fast inference, and low-latency TTS).

### A.1.1 LATENT ENCODER

The first convolutional layer of the latent encoder, followed by batch normalization, transforms a 228-dimensional mel spectrogram into hidden representations, preserving the sequence length and expanding the dimensionality to 512. The latent encoder employs 10 ConvNeXt blocks, with an intermediate dimension of 2048. The intermediate dimension refers to the hidden size between two consecutive $1 \times 1$ convolutional layers within each ConvNeXt block. A final linear layer, followed by layer normalization, projects the 512-dimensional output of the ConvNeXt blocks into a 24-dimensional latent space. All convolutional layers in the latent encoder use a kernel size of 7. In summary, the latent encoder compresses a 228-dimensional mel spectrogram into 24-dimensional latents while maintaining the original sequence length.

### A.1.2 LATENT DECODER

The latent decoder begins with a convolutional layer followed by batch normalization, transforming the 24-dimensional latents into hidden representations of size 512. It then processes these representations through 10 dilated ConvNeXt blocks, each with an intermediate dimension of 2048, followed by another batch normalization. The depthwise convolutional layers within these blocks use dilation rates of [1, 2, 4, 1, 2, 4, 1, 1, 1, 1]. Next, a convolutional layer with a kenel size of 3 converts the normalized output of the ConvNeXt blocks to hidden representations of dimension 2048. A final linear layer then projects these representations into frame-level outputs with 512 channels. These outputs are subsequently reshaped into a single-channel format, producing the final waveform output. The first convolutional layer and the depthwise convolutional layers within each ConvNeXt block use a kernel size of 7. Additionally, all convolutional layers in the latent decoder operate in a causal manner.

### A.1.3 DISCRIMINATOR

We adopt a lightweight version of multi-period discriminators (MPDs) introduced in HiFi-GAN (Kong et al., 2020). Each MPD consists of six convolutional layers with hidden sizes 16, 64, 256, 512, 512, and 1. The period settings remain the same as the original configuration: 2, 3, 5, 7, and 11. For multi-resolution discriminators (MRDs), log-scaled linear spectrograms serve as

Table 7: Configuration of convolutional layers in multi resolution discriminator.

| Layer | Input Channels | Output Channels | Kernel Size | Stride |
|---|---|---|---|---|
| Conv2D | 1 | 16 | (5, 5) | (1, 1) |
| Conv2D | 16 | 16 | (5, 5) | (2, 1) |
| Conv2D | 16 | 16 | (5, 5) | (2, 1) |
| Conv2D | 16 | 16 | (5, 5) | (2, 1) |
| Conv2D | 16 | 16 | (5, 5) | (1, 1) |
| Conv2D | 16 | 1 | (3, 3) | (1, 1) |

input, with three different FFT sizes: 512, 1024, and 2048. The hop sizes are set to one-quarter of the corresponding FFT size, while the window sizes equal to the FFT sizes. We use the Hann window function for spectral analysis. Each MRD consists of six convolutional layers, as detailed in Table 7.

## A.2 TEXT-TO-LATENT MODULE

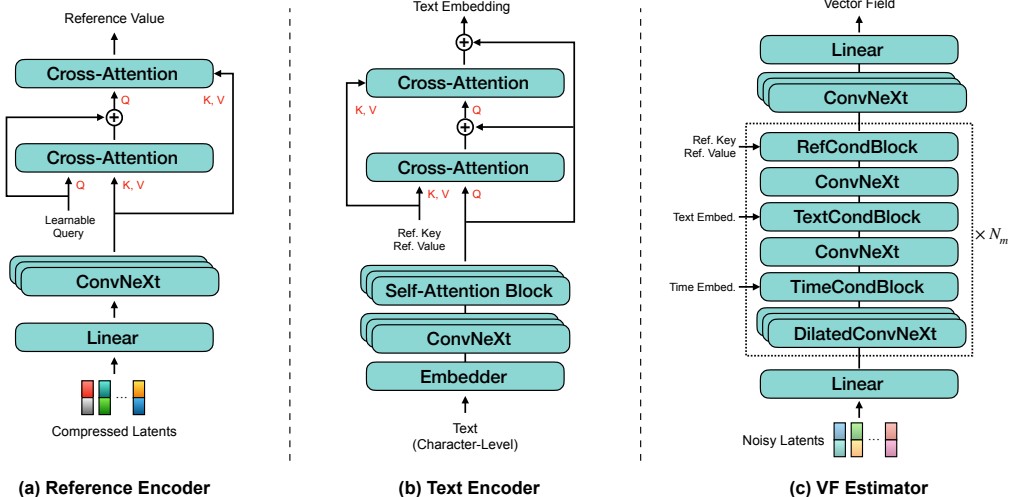

**(a) Reference Encoder**  **(b) Text Encoder**  **(c) VF Estimator**

Figure 4: Detailed architecture of reference encoder, text encoder, and VF estimator in text-to-latent module. Q, K, and V represent the inputs used to compute query, key, and value, respectively, in attention mechanism.

We illustrate the detailed architecture of the text-to-latent module in Fig. 4. Each component is carefully designed to operate efficiently with a simple architecture. Note that the text-to-latent module does not rely on any external pretrained models, G2P modules, or text-speech aligners.

### A.2.1 REFERENCE ENCODER

The reference encoder is composed of a linear layer, 6 ConvNeXt blocks, and 2 cross-attention layers. The linear layer transforms temporally compressed latents with a dimension of 144 to hidden representations with a dimension of 128. The kernel size and intermediate dimension of all ConvNeXt blocks are set to 5 and 512, respectively. In the cross-attention layers, three linear layers with the same input and output dimensions are used to generate query, key, and value. To obtain a fixed number of vectors (i.e., the reference value shown in Fig. 4 (a)) representing reference speech, 50 learnable vectors with a dimension of 128 are used in the first attention block.

### A.2.2 TEXT ENCODER

The text encoder consists of an embedder, 6 ConvNeXt blocks, 4 self-attention blocks, and 2 cross-attention layers. The embedder maps each character to a 128-dimensional vector with a simple

lookup table. The kernel size and intermediate dimension of ConvNeXt blocks are set to 5 and 512, respectively. The self-attention blocks follow the transformer encoder architecture, configured with 512 filter channels, 4 attention heads, and rotary position embedding. The cross-attention layers consist of three linear layers with same input and output dimensions, and the first cross-attention layer utilizes 50 learnable vectors (i.e., the reference key shown in Fig. 4 (b)), each with a dimension of 128. These 50 vectors are reused as keys in the VF estimator.

### A.2.3 VF ESTIMATOR

The first linear layer in the VF estimator maps 144-dimensional noisy latents to 256-dimensional hidden representations. The main block, highlighted with dotted lines in Fig. 4 (c), is composed of 4 dilated ConvNeXt blocks, 2 standard ConvNeXt blocks, TimeCondBlock, TextCondBlock, and RefCondBlock. Each ConvNeXt block has a kernel size of 5 and an intermediate dimension of 1024. The dilation rates for the four dilated ConvNeXt blocks are set to 1, 2, 4, and 8, respectively. TimeCondBlock employs a single linear layer to project a 64-dimensional time embedding onto the channel dimension of input and performs time conditioning via global addition. The time embedding is computed using the same method as in Grad-TTS (Popov et al., 2021). TextCondBlock and RefCondBlock employ a cross-attention mechanism to incorporate text and reference speech information, respectively. This structure is repeated four times ($N_m = 4$). Finally, 4 additional ConvNeXt blocks are applied, followed by a linear layer that maps the 256-dimensional representation back to a 144-dimensional output.

### A.3 DURATION PREDICTOR

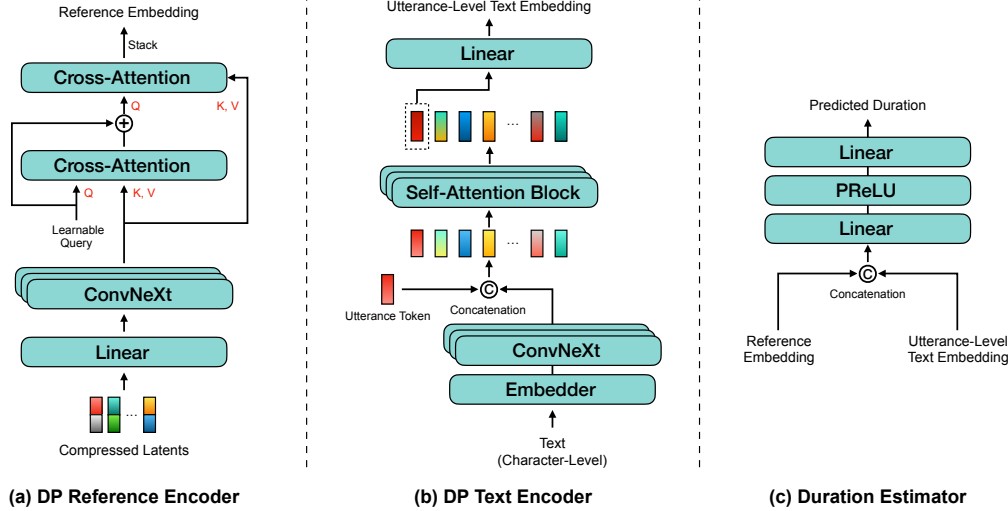

Figure 5: Detailed architecture of DP reference encoder, DP text encoder, and duration estimator in duration predictor.

We illustrate the detailed architecture of the duration predictor in Fig. 5. Since predicting total duration is simpler than phoneme-level duration prediction, we designed this module to be lightweight. Notably, it contains only about 0.5 million parameters.

### A.3.1 DP REFERENCE ENCODER

The DP reference encoder shares the same architecture as the reference encoder in the text-to-latent module but with different hyperparameter settings. It consists of a linear layer, 4 ConvNeXt blocks, and 2 cross-attention layers. The initial linear layer maps a 144-dimensional input to a 64-dimensional representation. Each ConvNeXt block has a kernel size of 5 and an intermediate dimension of 256. The cross-attention layers project inputs into 16-dimensional vectors and apply the attention mechanism. In the first cross-attention layer, queries are computed using eight learn-

able vectors. The final reference embedding is obtained by stacking the outputs along the channel dimension, resulting in a 64-dimensional vector.

### A.3.2   DP TEXT ENCODER

The DP text encoder comprises an embedder, 6 ConvNeXt blocks, 2 self-attention blocks, and a linear layer. The embedder converts character-level text input into 64-dimensional vectors. The kernel size and intermediate dimension of each ConvNeXt block are set to 5 and 256, respectively. A learnable 64-dimensional vector, referred to as the utterance token in Fig. 5 (b), is prepended to the output of the ConvNeXt blocks. The self-attention blocks have 256 filter channels, 2 attention heads, and incorporate rotary position embeddings. Finally, the first vector from the output of the last self-attention block passes through a linear layer with the same input and output dimension, producing an utterance-level text embedding.

### A.3.3   DURATION ESTIMATOR

The duration estimator consists of two linear layers with a PReLU activation. The first layer maintains the input and output dimensions at 164, while the second layer maps the output to a single scalar value. The outputs from the DP reference encoder and the DP text encoder are concatenated before being passing to the first linear layer, as shown in Fig. 5 (c).

## B   OPTIMIZATION DETAILS

### B.1   SPEECH AUTOENCODER

The training of the speech autoencoder is conducted within the framework of a Generative Adversarial Network (GAN) (Goodfellow et al., 2014). The two primary training objectives for the generator and discriminators are as follows:

$$\mathcal{L}_{\mathrm{G}} = \lambda_{\mathrm{recon}}\mathcal{L}_{\mathrm{recon}}(G) + \lambda_{\mathrm{adv}}\mathcal{L}_{\mathrm{adv}}(G; D) + \lambda_{\mathrm{fm}}\mathcal{L}_{\mathrm{fm}}(G; D), \tag{2}$$

$$\mathcal{L}_{\mathrm{D}} = \mathcal{L}_{\mathrm{adv}}(D; G), \tag{3}$$

where $G$ represents the generator, $D$ denotes the composite of discriminators. The reconstruction loss, $\mathcal{L}_{\mathrm{recon}}$, is computed using a spectral $L_1$ loss over multi-resolution mel spectrograms. Specifically, three separate mel spectrograms are generated using different FFT sizes: 1024, 2048, and 4096. These spectrograms are paired with corresponding mel band counts of 64, 128, and 128, respectively. Hop sizes are set to one-quarter of the corresponding FFT sizes. A Hann window is applied to each spectrogram, with the window size matching the respective FFT size used for that spectrogram. The adversarial losses, $\mathcal{L}_{\mathrm{adv}}$, are computed as follows:

$$\mathcal{L}_{\mathrm{adv}}(G; D) = \mathbb{E}_{x \sim p(x)} \left[ (D(G(x)) - 1)^2 \right], \tag{4}$$

$$\mathcal{L}_{\mathrm{adv}}(D; G) = \mathbb{E}_{x \sim p(x)} \left[ (D(G(x)) + 1)^2 + (D(x) - 1)^2 \right], \tag{5}$$

where $x$ denotes the ground truth audio and $G(x)$ represents the reconstructed adio. The feature matching loss, $\mathcal{L}_{\mathrm{fm}}$, is obtained by averaging the $L_1$ distances between intermediate features of each discriminator layer, derived from both real and generated speech:

$$\mathcal{L}_{\mathrm{fm}}(G; D) = \frac{1}{L} \sum_{l=1}^{L} \|\phi_l(G(x)) - \phi_l(x)\|_1, \tag{6}$$

where $L$ denotes the total number of layers in the discriminators and $\phi_l(\cdot)$ refers to the feature representations obtained from the $l$-th discriminator layer.

---

**Algorithm 1** Training with Context-Sharing Batch Expansion

---

**Require:** Diffusion model $f_\theta$, condition encoder $g_\phi$, mini-batch $\{(x_i, c_i)\}_{i=1}^B$, expansion factor $K_e$
 1: Encode the conditional variables: $\{\bar{c}_i\}_{i=1}^B \leftarrow g_\phi(\{c_i\}_{i=1}^B)$
 2: Initialize expanded batch $\mathcal{B}_{\text{exp}} \leftarrow \emptyset$
 3: **for** $i = 1$ to $B$ **do**
 4:     **for** $k = 1$ to $K_e$ **do**
 5:         Sample noise $\epsilon_i^k$, timestep $t_i^k$
 6:         $\tilde{x}_i^k \leftarrow \text{ForwardProcess}(x_i, \epsilon_i^k, t_i^k)$           ▷ Perturb the input with noise at timestep $t$
 7:         Append $(\tilde{x}_i^k, \bar{c}_i, t_i^k)$ to $\mathcal{B}_{\text{exp}}$     ▷ Encoded condition $\bar{c}_i$ is shared across $K_e$ samples
 8:     **end for**
 9: **end for**
10: Optimize $f_\theta$ using $\mathcal{B}_{\text{exp}}$

---

Table 8: Performance comparison across different numbers of function evaluations. Best results are highlighted in bold, and second-best are underlined.

|      | NFE | RTF ↓ | WER ↓ | SIM ↑ | NISQA ↑ |
|------|-----|-------|-------|-------|---------|
| GT   | -   | -     | 2.181 | $0.677 \pm 0.006$ | $4.070 \pm 0.029$ |
| Ours | 4   | **0.006** | 11.43 | $0.335 \pm 0.003$ | $2.623 \pm 0.020$ |
|      | 8   | 0.011 | 2.818 | $0.472 \pm 0.003$ | $3.916 \pm 0.014$ |
|      | 16  | 0.019 | 2.679 | **$0.476 \pm 0.003$** | $3.994 \pm 0.014$ |
|      | 32  | 0.037 | **2.639** | $0.472 \pm 0.003$ | $4.033 \pm 0.014$ |
|      | 64  | 0.071 | 2.705 | $0.470 \pm 0.003$ | $4.060 \pm 0.014$ |
|      | 128 | 0.140 | 2.693 | $0.468 \pm 0.003$ | **$4.070 \pm 0.014$** |

## C  ALGORITHM FOR CONTEXT-SHARING BATCH EXPANSION

We provide a pseudo-algorithm for context-sharing batch expansion in Algorithm 1.

## D  ADDITIONAL EXPERIMENTAL RESULTS

### D.1  TRADE-OFF BETWEEN THE NUMBER OF FUNCTION EVALUATIONS AND SYNTHESIS QUALITY

SupertonicTTS synthesizes speech through Euler's method during inference. By adjusting the number of function evaluations (NFE), we can balance the trade-off between synthesis speed and quality. To quantify this relationship, we generated five samples per utterance from the *LS-clean* by varying NFE values while keeping a CFG coefficient to 3. These samples were then evaluated in terms of RTF, WER, NISQA score (Mittag et al., 2021), and speaker similarity (SIM). SIM was calculated as the cosine similarity between speaker embeddings from the generated speech and the corresponding 3-second reference, using the WavLM-TDNN model (Chen et al., 2022).

The evaluation results, presented in Table 8, demonstrate a general trend that increasing NFE leads to improvements in the WER, SIM, and NISQA scores. Specifically, the best scores for WER and SIM were obtained at NFE values of 32 and 16, respectively. This suggests that NFE values exceeding 32 effectively capture intelligibility, prosodic naturalness, and speaker identity. Meanwhile, NISQA scores exhibited a consistent positive correlation with NFE, suggesting that higher NFE values result in enhanced audio fidelity. However, this improvement comes at the cost of increased processing time, as reflected in the RTF. Given this trade-off, we select NFE $= 32$ as the optimal balance between quality and efficiency.

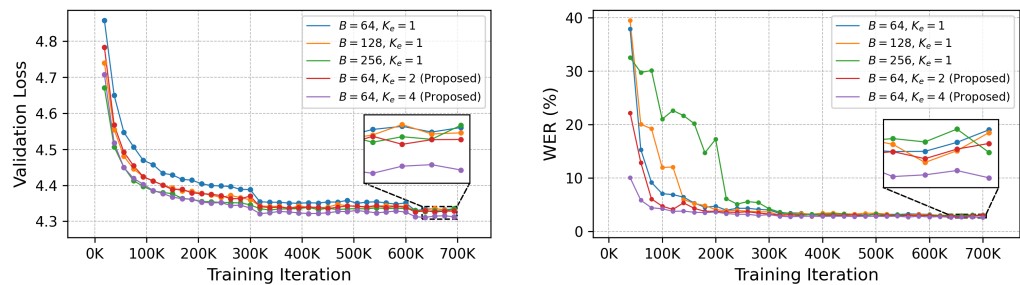

Figure 6: Model performance tracked throughout the entire training process.

## D.2 EXPERIMENTAL RESULTS ON EVALUATION OF CONTEXT-SHARING BATCH EXPANSION

Fig. 6 presents validation loss and ASR results throughout the entire training process. Increasing the expansion factor $K_e$ consistently accelerates convergence for both validation loss and WER. In contrast, while increasing the batch size $B$ helps reduce validation loss, it slows the convergence of WER. This indicates that simply increasing the batch size is not sufficient for achieving accurate text-speech alignment. Additionally, this experiment demonstrates that the proposed batch expansion algorithm not only accelerates loss convergence but also alleviates issues such as word skipping, repetition, and mispronunciation.

## E SUBJECTIVE LISTENING TEST

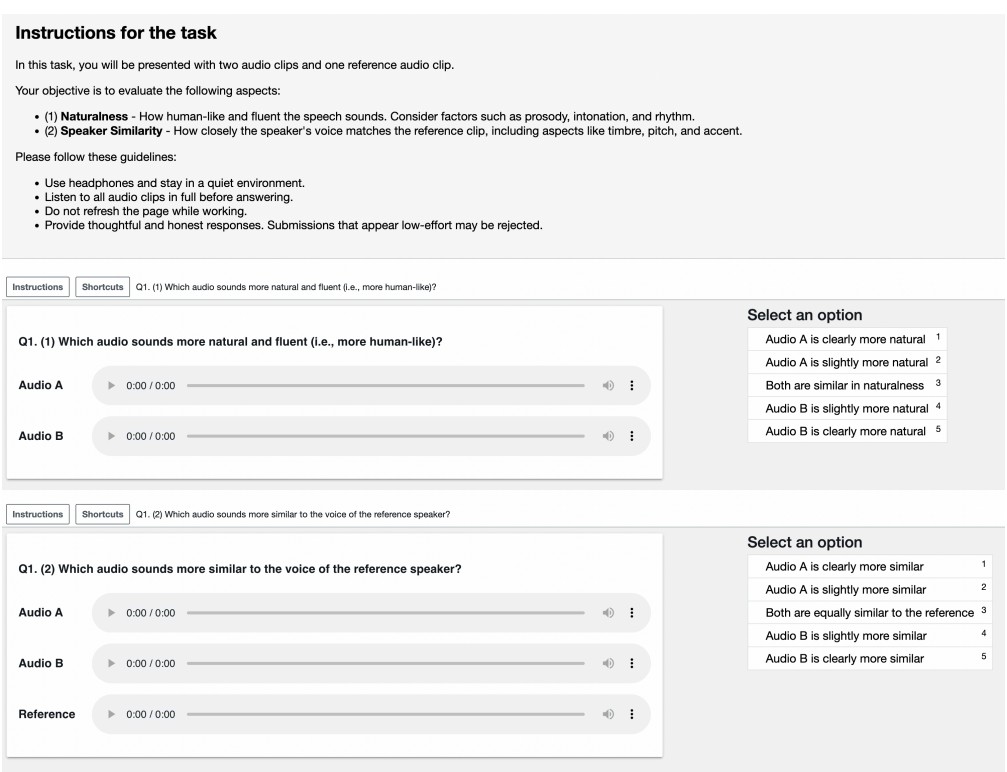

Figure 7: Example of subjective evaluation page.

Fig. 7 illustrates the survey interface presented to the evaluation participants. All audio files were resampled to $16\,\mathrm{kHz}$ before the subjective listening tests.

Table 9: Public datasets used to train the speech autoencoder.

| Dataset | URL |
|---------|-----|
| AISHELL-3 (Shi et al., 2021) | https://www.openslr.org/93/ |
| Att-HACK (Moine & Obin, 2020) | https://www.openslr.org/88/ |
| DAPS (Mysore, 2015) | https://zenodo.org/records/4660670 |
| EARS (Richter et al., 2024) | https://sp-uhh.github.io/ears_dataset/ |
| Hi-Fi TTS (Bakhturina et al., 2021) | https://www.openslr.org/109/ |
| JSUT (Sonobe et al., 2017) | https://sites.google.com/site/shinnosuketakamichi/publication/jsut |
| LibriTTS (Zen et al., 2019) | https://www.openslr.org/60/ |
| PTDB-TUG (Pirker et al., 2011) | https://www.spsc.tugraz.at/databases-and-tools |
| RAVDESS (Livingstone & Russo, 2018) | https://zenodo.org/record/1188976 |
| VCTK (Yamagishi et al., 2019) | https://datashare.ed.ac.uk/handle/10283/2950 |
| SLR32 (van Niekerk et al., 2017) | https://www.openslr.org/32/ |
| SLR41 (Sodimana et al., 2018) | https://www.openslr.org/41/ |
| SLR42 (Sodimana et al., 2018) | https://www.openslr.org/42/ |
| SLR43 (Sodimana et al., 2018) | https://www.openslr.org/43/ |
| SLR44 (Sodimana et al., 2018) | https://www.openslr.org/44/ |
| SLR61 (Guevara-Rukoz et al., 2020) | https://www.openslr.org/61/ |
| SLR63 (He et al., 2020) | https://www.openslr.org/63/ |
| SLR64 (He et al., 2020) | https://www.openslr.org/64/ |
| SLR65 (He et al., 2020) | https://www.openslr.org/65/ |
| SLR66 (He et al., 2020) | https://www.openslr.org/66/ |
| SLR69 (Kjartansson et al., 2020) | https://www.openslr.org/69/ |
| SLR70 | https://www.openslr.org/70/ |
| SLR71 (Guevara-Rukoz et al., 2020) | https://www.openslr.org/71/ |
| SLR72 (Guevara-Rukoz et al., 2020) | https://www.openslr.org/72/ |
| SLR73 (Guevara-Rukoz et al., 2020) | https://www.openslr.org/73/ |
| SLR74 (Guevara-Rukoz et al., 2020) | https://www.openslr.org/74/ |
| SLR75 (Guevara-Rukoz et al., 2020) | https://www.openslr.org/75/ |
| SLR76 (Kjartansson et al., 2020) | https://www.openslr.org/76/ |
| SLR77 (Kjartansson et al., 2020) | https://www.openslr.org/77/ |
| SLR78 (He et al., 2020) | https://www.openslr.org/78/ |
| SLR79 (He et al., 2020) | https://www.openslr.org/79/ |
| SLR80 (Kjartansson et al., 2020) | https://www.openslr.org/80/ |
| SLR83 (Demirsahin et al., 2020) | https://www.openslr.org/83/ |
| SLR86 (Gutkin et al., 2020) | https://www.openslr.org/86/ |

## F PUBLIC DATASETS

Table 9 provides a list of public datasets used for training the speech autoencoder.

## G LIMITATIONS AND FUTURE WORK

While we showed that SupertonicTTS is scalable and efficient, certain limitations offer avenues for future research. Firstly, while SupertonicTTS is designed for linguistic scalability, particularly with its direct raw character input, the experiments in this paper were conducted exclusively on English. Future work could involve multilingual experiments to further demonstrate its adaptability and advantages across diverse languages. Secondly, although SupertonicTTS achieves fast inference with 32 function evaluations, there is room for further speed improvements. Incorporating advanced distillation techniques could substantially reduce the number of required function evaluations, thereby enhancing inference speed even further. Lastly, the decoding performance of the speech autoencoder sets an upper bound on the overall quality of synthesized speech. Given that the text-to-latent module accounts for the majority of computational cost, developing a more expressive speech autoencoder could improve output quality with minimal impact on overall efficiency.

## H BROADER IMPACTS

We believe that SupertonicTTS can yield several positive societal impacts. For instance, it can facilitate voice-based human-computer interaction and help democratize audio content creation by lowering technical and computational resource hurdles. Also, it can enhance accessibility to digital

information for individuals with visual impairments or reading difficulties. However, the progress in realistic and accessible voice synthesis presents potential negative consequences. There may be a risk of misuse of synthetic voice for disinformation or fraud, particularly with zero-shot voice cloning capabilities. These challenges could be addressed with robust synthetic speech detection, audio watermarking techniques, and ethical guidelines for the use and deployment of voice synthesis technologies.

## I LARGE LANGUAGE MODELS (LLMS) IN PAPER WRITING

We used LLMs for assistance with phrasing, grammar, and overall clarity of sections within this manuscript. All technical content, research contributions, and original ideas are the sole work of the authors.

