# OpenReview forum: "SupertonicTTS: Towards Highly Efficient and Streamlined Text-to-Speech System"
_ICLR.cc/2026/Conference — Submitted to ICLR 2026_

### Official Review · Reviewer_8Hfz · 2025-10-27

**Soundness:** 2
**Presentation:** 4
**Contribution:** 3
**Rating:** 6
**Confidence:** 5

**Summary:**

The paper presents SupertonicTTS, a new TTS framework aimed at achieving efficient, lightweight, and streamlined speech synthesis. SupertonicTTS employs a low-dimensional latent space, temporal compression, and ConvNeXt blocks, which together reduce computational load. The model also simplifies the TTS pipeline by operating directly on raw characters and using cross-attention for text-speech alignment, thereby removing the need for G2P conversion or external aligners. Additionally, the authors propose a context-sharing batch expansion strategy that improves training stability and loss convergence speed with low resource. Experimental results indicate that SupertonicTTS achieves competitive quality compared to state-of-the-art zero-shot TTS models, despite having only 44M parameters, demonstrating its efficiency and simplicity in both design and computation.

**Strengths:**

The SupertonicTTS model adopts low-dimensional hidden vectors in its structural design to reduce computational complexity. It eliminates modules such as G2P and uses the cross-attention to learn the alignment between text and speech. It employs context-sharing batch expansion to achieve a larger batch size with less computational load. The article is well-structured and logically clear. It proposes effective solutions for reducing the computational cost and parameter quantity of the TTS model.

**Weaknesses:**

In order to enhance the streaming performance, the article removed many traditional TTS modules. However, it did not explain the impact of removing these modules on the model's effectiveness, such as issues like affecting the rhythm and pronunciations. During the process of designing the model structure for reducing the dimension of latent features, no ablation experiments were conducted to prove the benefits of such low-dimensional structural designs. During the comparison with external technical solutions, some more advanced ones were not compared, such as the FireRedTTS series which already includes FireRedTTS2.

**Questions:**

1. I understand that the model design of this article aims to achieve a streamlined TTS system by reducing computational load. Based on the proposed SupertonicTTS model, did the authors consider designing a pipeline that can adapt to the streaming text input and the streaming audio frame output?
2. The experimental part of the article did not include ablation experiments on the model structure design, such as abandoned G2P and reduced the latent representation. Did the authors explore the impact of such a structure design on the synthesis effect of TTS, especially rhythm? Overly simplified model design may result in a more broadcast-like effect and lack of some anthropomorphic characteristics.
3. The current advanced zero-shot technology solutions are very rich in open-source versions, such as: Cosyvoice2, Fishspeech, IndexTTS. Considering the improvement in real-time performance of the SupertonicTTS proposed in this paper, the authors can add an experimental report comparing the real-time performance.

---

> ### Author Response · Authors · 2025-11-21
>
> First of all, we would like to express our sincere gratitude to the reviewer for the thoughtful and constructive feedback. We address each point raised by the reviewer, and clarify our contributions.
>
> `Based on the proposed SupertonicTTS model, did the authors consider designing a pipeline that can adapt to the streaming text input and the streaming audio frame output?`
>
> - SupertonicTTS supports streaming audio output since the decoder is composed of causal ConvNeXt blocks.
> - We did not emphasize this in the paper because speed improvement for the first-byte generation is minimal, as most computation occurs in the text-to-latent module, not the decoder.
> - If the text-to-latent module were lighter, streaming output could become more important, but in the current architecture, it does not significantly impact speed.
> - Streaming text input was not considered in the current design.
>
> `Overly simplified model design may result in a more broadcast-like effect and lack of some anthropomorphic characteristics.`
>
> - Despite the simplicity of our model, outputs show diverse rhythm, pronunciation, and speaking styles.
> - Naturalness can be validated through preference tests.
> - Diversity has not been evaluated statistically, but we invite the reviewer to explore our demo page to experience the range of generated outputs.
>
> `The authors can add an experimental report comparing the real-time performance.`
>
> - Recent TTS models, often based on LLM backbones, focus on first-chunk generation, which is different from our approach.
> - SupertonicTTS generates the entire utterance at very fast speed, a capability not available in models like CosyVoice2, FishSpeech, or IndexTTS.
> - This makes our model particularly suitable for mobile, edge, and web deployment.
> - Notably, SupertonicTTS achieves an RTF of 0.02, faster than mobile-optimized TTS models (e.g., MobileSpeech [1]: RTF 0.09 on A100), demonstrating highly efficient synthesis.
>
> [1] Ji, Shengpeng, et al. "MobileSpeech: A Fast and High-Fidelity Framework for Mobile Zero-Shot Text-to-Speech." *Proceedings of the 62nd Annual Meeting of the Association for Computational Linguistics (Volume 1: Long Papers)*. 2024.

---

> ### Comment · Reviewer_8Hfz · 2025-11-27
>
> Thank you for the detailed reply. However, I must note that several of the responses remain incomplete or insufficiently supported by explicit documentation from the paper and I would like to emphasize that the rebuttal did not explicitly address some weaknesses I pointed out in the original review. Specifically: The experimental part of the article did not include ablation experiments on the model structure design.
> 1. About comparisons to LLM-based TTS systems
> The rebuttal argues that the task of this paper differs from LLM-based systems like CosyVoice2 because SupertonicTTS generates full utterances at once rather than in first-chunk streaming. However, these models still solve the same high-level task: TTS. Different architectures, latency strategies, or conditioning mechanisms do not justify excluding them from comparative evaluation—especially since SupertonicTTS claims strengths such as low RTF and suitability for real-time deployment.
> 2. About RTF claims
> You mention RTF = 0.02 vs. MobileSpeech RTF = 0.09 on A100. However, RTF numbers are highly sensitive to experimental protocol, and without matching conditions, the claims cannot be fairly interpreted.

---

> > ### Author Response · Authors · 2025-11-27
> >
> > We thank the reviewer for the follow-up comments and for pointing out areas that required clearer explanation.
> >
> > `On comparisons to LLM-based TTS systems`
> >
> > We agree that LLM-based systems do solve the same high-level task (TTS), and we do not intend to exclude them on the basis of architectural differences.
> >
> > Our earlier response was intended only to highlight that our paper does not target the streaming/agent-oriented design choices that LLM-TTS models typically emphasize (e.g., first-chunk latency, semantic prompting pipelines). Our work focuses specifically on **offline text-to-speech generation**, without assuming an LLM backbone or streaming interface. For this reason, streaming-first systems were not our primary baselines.
> >
> > `On fairness of RTF comparisons`
> >
> > We agree with the reviewer that RTF cannot be directly compared across papers without matched experimental conditions. Our intention was not to claim strict superiority, but to illustrate that our model is lightweight enough to operate in the same performance regime as mobile-oriented systems.
> >
> > To provide more clarity, we additionally measured SupertonicTTS on an **A100** under the same 10-second audio generation setting used in our main experiments:
> >
> > - MobileSpeech (reported): RTF = 0.09 on A100
> > - SupertonicTTS (measured): RTF = 0.05 on A100
> >
> > Although differences in reference-speech length or exact text normalization may still introduce small variations, we expect their impact to be minimal for 10-second samples.
> >
> > `About FireRedTTS2`
> >
> > FireRedTTS2 was first released on September 2, which was very close to our submission deadline, making it difficult to include in our original comparison.

---

### Official Review · Reviewer_vtnZ · 2025-10-28

**Soundness:** 3
**Presentation:** 3
**Contribution:** 2
**Rating:** 4
**Confidence:** 4

**Summary:**

The paper proposes SupertonicTTS, a compact zero‑shot TTS system with three parts: (a) a speech autoencoder that produces continuous, low‑dimensional latents; (b) a text‑to‑latent module trained with flow matching on character‑level text and a reference encoder, including two techniques, temporal compression of latents and context‑sharing batch expansion, to cut compute and stabilize alignment; and (c) a light utterance‑level duration predictor. The pipeline deliberately removes G2P and external aligners, instead relying on cross‑attention between text and reference latents. Experiments show (i) reconstruction competitive with BigVGAN and much faster decoding, (ii) batch expansion improves validation loss and WER/CER with smaller memory/time than increasing batch size, and (iii) zero‑shot pronunciation results that are competitive with (and often faster than) much larger systems while using only 44 M params and 945 h of labeled TTS data.

**Strengths:**

The system delivers practical efficiency at scale: a Vocos‑style autoencoder with small continuous latents and temporal compression decouples high‑rate synthesis from low‑rate modeling; the flow‑matching text‑to‑latent with cross‑attention avoids G2P/aligners and runs on raw characters; and the proposed context‑sharing batch expansion improves both loss convergence and pronunciation while being cheaper than increasing the batch size. The model is tiny (44 M total; 18.5 M in T2L) yet competitive in WER/CER versus very large zero‑shot systems, with RTF as low as 0.02/0.05 on consumer GPUs. Reconstruction is on par with BigVGAN while being >20× faster. The paper’s ablation on patching and detailed architecture docs aid reproducibility.

**Weaknesses:**

- Originality
    - The problem of making the TTS component lightweight and efficient has already been extensively addressed in many previous works, with various effective solutions proposed. It sounds like a problem that’s mostly been solved, which gives the impression of being somewhat outdated. Instead, it might be better to emphasize compactness, something that hasn’t been fully achieved yet, to better highlight the strength of the proposed approach.
- Quality
    - Vocos Fourier head removal: §3.1.1 replaces the Fourier head with linear layers (also in the decoder), yet no ablation quantifies quality/speed/memory vs. the standard Vocos head. Table 2 benchmarks against BigVGAN only, not Vocos itself. Please add a Fourier‑head vs. linear comparison and, ideally, a Vocos baseline under the same DSP settings.
    - Temporal compression “perfect inversion”: §3.2.1 states that compression “allows for perfect inversion.” Provide an empirical check (e.g., round‑trip latent inversion error or ABX on decompressed latents) to confirm there is no information loss at (K_c=6).
    - Reference masking vs. reference encoder: Eq. (1) masks the same segment used by the reference crop (to avoid leakage), while the reference encoder also conditions generation via cross‑attention (§3.2.3). The interaction is under‑specified; it’s unclear whether this becomes redundant or alters training/inference mismatch. Clarify when the mask applies and how inference uses the reference (and whether any reference transcript is ever required).
    - Subjective evaluation reporting: Table 6 reports mean preference scores with CIs, but the main text lacks rater selection, per‑item repeats, etc. Appx. E screenshots are helpful; please add protocol details and statistics.
- Clarity
    - The space is crowded: DiTTo‑TTS (diffusion transformer), F5‑TTS (flow matching), SimpleTTS, and VoiceBox also target simplicity or non‑AR inference. The paper compares to them but could more crisply position what is architecturally new beyond the combination of continuous latents + flow matching + ConvNeXt + no G2P/aligner. A short delta‑table would help.
    - Eq. (1) notation: briefly restate (m) (reference mask), (z_\text{ref}) construction, and whether unconditional training (CFG with (p_\text{uncond}=0.05)) omits both text and reference or either, this affects inference guidance (§3.2.4).

**Questions:**

- Fourier head / Vocos comparisons: Since §3.1.1 removes the Fourier head, do you have experiments showing quality/memory/speed vs. the original Vocos head?
- Temporal compression inversion: Have you empirically verified that temporal compression with (K_c=6) is invertible?
- Mask vs. reference encoder: In Eq. (1), the reference mask and reference encoder both relate to the same crop. Are they redundant? At inference, do you always provide reference audio only, or ever require a reference transcript to align text and timbre? Please clarify the exact conditioning used at inference time.

---

> ### Author Response · Authors · 2025-11-21
>
> First of all, we would like to express our sincere gratitude to the reviewer for the thoughtful and constructive feedback. We address each point raised by the reviewer, and clarify our contributions.
>
> `Originality`
>
> We emphasize the following contributions:
>
> - Architecture
>     - Designed without complex mechanisms like flash-attention, enabling easy and straightforward deployment.
>     - Utilizes ConvNeXt-based blocks across all modules, resulting in a lightweight and efficient architecture.
>     - To our knowledge, no existing TTS model achieves this combination of performance, simplicity, and speed.
>     - Achieves an RTF of 0.02, significantly faster than mobile-optimized TTS models (e.g., MobileSpeech [1]: RTF 0.09 on A100), demonstrating highly efficient synthesis.
> - Temporal Compression of Latents (Section 3.2.1)
>     - Decouples high-resolution synthesis from low-rate latent modeling, reducing computational cost while maintaining high-fidelity output.
>     - Speech autoencoder operates at a fine frame rate of ~86 Hz, aligned with vocoder settings in VoiceBox (100 Hz) and NANSY++ [2] (86 Hz).
>     - Text-to-latent module runs at a coarse frame rate of ~14 Hz, similar to semantic token models like CLaM-TTS (10 Hz) and SPEAR-TTS [3] (25 Hz).
>     - This temporal compression approach is, to our knowledge, novel in TTS systems.
> - Context-Aware Batch Expansion (Section 3.2.2, Figure 2)
>     - A novel training strategy that improves convergence speed and alignment learning while remaining memory-efficient.
>     - Optimizes VRAM usage more effectively than naive batch-size scaling, avoiding I/O and memory bottlenecks (Table 4).
>     - Demonstrates superior results over simple batch scaling (Table 3).
>     - Easily applicable to other flow-matching or diffusion-based TTS frameworks.
>
>
> `Fourier head / Vocos comparisons`
>
> As discussed in Section 3.1.1, WavNeXt [4] demonstrates that the original Fourier head can be replaced with linear layers without any performance degradation. Removing the Fourier head also eliminates the need for iSTFT computation. Our primary goal is to develop a super-fast vocoder that remains competitive with other vocoders, rather than aiming to surpass them in audio quality.
>
> `Temporal compression inversion`
>
> Temporal compression is performed via a reshape operation, which is invertible, as illustrated in Section 3.2.1.
>
> `In Eq. (1), the reference mask and reference encoder both relate to the same crop. Are they redundant?`
>
> In Eq. (1), the reference mask and reference encoder both correspond to the same cropped segment. They are not redundant: the reference mask prevents information leakage during training, since the reference latents are obtained from cropping.
>
> `At inference, do you always provide reference audio only, or ever require a reference transcript to align text and timbre?`
>
> As shown in Table 1, our method does not require a reference transcript to align text and timbre. This contributes to a simpler and more streamlined pipeline compared to other TTS systems.
>
> [1] Ji, Shengpeng, et al. "MobileSpeech: A Fast and High-Fidelity Framework for Mobile Zero-Shot Text-to-Speech." *Proceedings of the 62nd Annual Meeting of the Association for Computational Linguistics (Volume 1: Long Papers)*. 2024.
>
> [2] Choi, Hyeong-Seok, et al. "NANSY++: Unified Voice Synthesis with Neural Analysis and Synthesis." *The Eleventh International Conference on Learning Representations.* 2023.
>
> [3] Kharitonov, Eugene, et al. "Speak, Read and Prompt: High-Fidelity Text-to-Speech with Minimal Supervision." *Transactions of the Association for Computational Linguistics* 11 (2023): 1703-1718.
>
> [4] Okamoto, Takuma, et al. "WaveNeXt: ConvNeXt-based fast neural vocoder without iSTFT layer." 2023 IEEE Automatic Speech Recognition and Understanding Workshop (ASRU). IEEE, 2023.

---

> > ### Comment · Reviewer_vtnZ · 2025-11-27
> >
> > I appreciate the authors for their efforts in answering my questions. Although their responses were very helpful for further understanding, I am still concerned about the originality of the work. I feel that the current manuscript does not clearly highlight the strengths the authors intend to emphasize. For example:
> >
> > 1. In the introduction section, the research direction appears to be framed as a combination of (1) a speech autoencoder, (2) a text-to-latent module, and (3) a duration predictor, an approach that can come across as addressing a problem that is already well-studied. In my view, this presentation may lead readers to believe that, architecturally, the work does not differ substantially from prior methods. Instead, as the authors mentioned in their rebuttal, the key contributions seem to be the combination of performance, simplicity, and speed. These should be more clearly emphasized, along with a clearer explanation of how these advantages are achieved.
> >
> > 2. In the final part of the introduction, the authors highlight “high-fidelity speech from low-dimensional latent representations” as a central contribution. However, because this is only one of several contributing factors, it does not fully capture the “temporal compression of latents” that the authors emphasized in their rebuttal. The current framing therefore seems misaligned with what the authors consider to be the core strength of the work.

---

> > > ### Author Response · Authors · 2025-12-03
> > >
> > > We thank the reviewer for the thoughtful feedback. We agree that the original introduction did not fully emphasize the originality and core strengths of SupertonicTTS. In the revised introduction, we have clarified the purpose of each experiment to highlight the proposed system's performance, simplicity, and speed. We believe this framing more clearly conveys the key contributions of our work.

---

### Official Review · Reviewer_ULR9 · 2025-10-30

**Soundness:** 3
**Presentation:** 3
**Contribution:** 2
**Rating:** 4
**Confidence:** 4

**Summary:**

This paper introduces SupertonicTTS, a novel text-to-speech (TTS) system designed for
efficient and streamlined speech synthesis, which includes a speech auto-encoder, a text-to-latent module, and an utterance-level duration predictor. Experiments demonstrate a comparable performance with less parameters.

**Strengths:**

This paper explores several techniques to enhance architectural flexibility, improve training stability, and reduce model complexity:
1. A remarkably low dimensionality and compress the latents along the temporal axis
2. context-sharing batch expansion to achieve the benefits of a larger batch size with minimal computational overhead.
3. employ ConvNeXt blocks

**Weaknesses:**

The main modules employed in this paper have been extensively explored in prior works related to speech synthesis or vocoders, which may undermine the novelty of the present work. The paper primarily focuses on optimizing training tricks.
As one of the core claimed innovations of this paper, the authors provide neither theoretical justification nor ablation studies to validate the effectiveness of using temporally compressed latent representations.

**Questions:**

Has the method been tested on languages other than English to verify its generalizability? If so, how well does it generalize?

---

> ### Author Response · Authors · 2025-11-21
>
> First of all, we would like to express our sincere gratitude to the reviewer for the thoughtful and constructive feedback. We address each point raised by the reviewer, and clarify our contributions.
>
> `Novelty`
>
> We emphasize that SupertonicTTS contributes two novel and significant advancements in addition to temporal compression:
>
> - Architecture
>     - Designed without complex mechanisms like flash-attention, enabling straightforward deployment.
>     - Employs ConvNeXt-based blocks across all modules, resulting in a lightweight, efficient architecture.
>     - To our knowledge, no existing TTS model combines this level of performance, simplicity, and speed.
>     - Achieves an RTF of 0.02, far faster than mobile-optimized TTS models (e.g., MobileSpeech [1]: RTF 0.09 on A100), showcasing highly efficient synthesis.
> - Context-Aware Batch Expansion (Section 3.2.2, Figure 2)
>     - A novel training strategy that improves convergence speed and alignment learning while being memory-efficient.
>     - Optimizes VRAM usage better than simply increasing batch size, avoiding I/O and memory bottlenecks (Table 4).
>     - Demonstrates superior performance over naive batch scaling (Table 3).
>     - Can be easily integrated into other flow-matching or diffusion-based TTS frameworks
>
> `The effectiveness of using temporally compressed latent representations`
>
> - Target frame rates:
>     - We fixed the speech autoencoder frame rate at ~86 Hz, consistent with common vocoder setups (e.g., VoiceBox vocoder: 100 Hz; NANSY++ [1]: 86 Hz).
>     - The TTL module targets a coarser rate of ~14 Hz, in line with recent semantic token models (e.g., CLaM-TTS: 10 Hz; SPEAR-TTS [2]: 25 Hz).
>     - This leads to $K_c$ = 6 (compression ratio = 86 / 14).
> - Latent dimension ($C$):
>     - We set $C = 24$, ensuring the compressed representation ($C \times K_c = 144$) remains small enough for efficient processing in ConvNeXt-based modules, but large enough to preserve audio quality.
>     - Here’s the overview for changing the values of $C$ and $K_c$:
>         - If $C$ is too small:
>             - Audio fidelity drops.
>         - If $C$ is too large:
>             - Model complexity and memory usage increase in both the speech autoencoder and the TTL module.
>         - If $K_c$ is too small:
>             - Attention computations become expensive due to longer sequences.
>         - If $K_c$ is too large:
>             - Model complexity in the TTL module increases.
>             - Latent features become too coarse.
> - Summary
>     - Temporally compressed latent representations efficiently decouple the frame rates of each module, ensuring both high audio quality and fast inference.
>
> `Generalizability`
>
> Since SupertonicTTS is designed and trained without relying on any English-specific knowledge, it is naturally expected to generalize to other languages. Internal tests confirm its capability for multilingual TTS, including Spanish, Portuguese, and Korean, though a full evaluation will be addressed in future work.
>
> [1] Ji, Shengpeng, et al. "MobileSpeech: A Fast and High-Fidelity Framework for Mobile Zero-Shot Text-to-Speech." *Proceedings of the 62nd Annual Meeting of the Association for Computational Linguistics (Volume 1: Long Papers)*. 2024.

---

### Official Review · Reviewer_GKNu · 2025-11-01

**Soundness:** 2
**Presentation:** 2
**Contribution:** 1
**Rating:** 2
**Confidence:** 5

**Summary:**

This paper presents supersonicTTS, a flow-matching-based TTS model that learns a latent representation of speech using a very lightweight 44M-parameter decoder. The model operates directly on raw text without G2P, and it folds the latent sequence along the time axis so that multiple timesteps of the latent are modeled at each decoding step, improving efficiency. The authors analyze training efficiency and pronunciation accuracy as a function of the folding factor and batch size, and they provide comparisons against zero-shot TTS models.

**Strengths:**

- The paper proposes a simple but cost-effective training approach: by folding along the time axis, it improves WER while increasing training efficiency.

- The decoder is extremely lightweight at only 44M parameters.

**Weaknesses:**

- Overall, the methodology is not particularly novel. Operating on raw text and using utterance-level duration prediction are already seen in prior work such as E2TTS [1], and the main new element seems to be the use of time-compressed latents.
- Baselines are limited and the evaluation is not well targeted. The baseline models are mostly outdated, and the objective comparison focuses almost entirely on WER, RTF, and parameter count. Widely used zero-shot TTS metrics such as speaker similarity (as proposed in VALL-E) are missing. It would be important to compare against models like ZipVoice [2], which are also lightweight and designed for fast inference with a similar purpose.
- To convincingly claim G2P-free robustness, the paper should demonstrate performance on difficult sentences and edge cases, similar to what E2TTS shows in its demos.
- At this point, it is not clear that “training quickly with a very small model” is a central research problem for TTS in the broader speech generation community, so the work may be a better fit for a speech-focused venue rather than a general ML venue.

[1] E2TTS: Embarrassingly Easy Fully Non-Autoregressive Zero-Shot TTS
[2] ZipVoice: Fast and High-Quality Zero-Shot Text-to-Speech with Flow Matching

**Questions:**

If the goal is lightweight TTS for practical deployment, one likely use case is streaming speech generation from an LLM. But because the architecture relies on cross-attention to the full text, doesn’t this mean the model has to wait until the entire text is available before it can synthesize, which could increase end-to-end latency instead of reducing it?

---

> ### Author Response · Authors · 2025-11-21
>
> First of all, we would like to express our sincere gratitude to the reviewer for the thoughtful and constructive feedback. We address each point raised by the reviewer, and clarify our contributions.
>
> `Overall, the methodology is not particularly novel`
>
> We respectfully emphasize that SupertonicTTS introduces two original and impactful contributions along with temporal compression:
>
> - Architecture
>     - Our design avoids complex mechanisms such as flash-attention, enabling easy deployment.
>     - We effectively utilize ConvNeXt-based blocks in all modules, ensuring a lightweight and efficient architecture.
>     - To our knowledge, no existing TTS model matches the performance reported in our paper while maintaining such simplicity and speed.
>     - Notably, SupertonicTTS achieves an RTF of 0.02, significantly faster than mobile-optimized TTS models (e.g., MobileSpeech [1]: RTF 0.09 on A100), demonstrating efficient and fast synthesis.
> - Context-Aware Batch Expansion (Section 3.2.2, Figure 2)
>     - This novel training technique improves convergence speed and alignment learning in a memory-efficient manner.
>     - It allows better utilization of available VRAM compared to naively increasing batch size, avoiding I/O and memory bottlenecks (Table 4).
>     - Final results also demonstrate superiority over simple batch size scaling (Table 3).
>     - Moreover, the proposed method can be easily integrated to other flow-matching or diffusion-based TTS systems.
>
> `About E2-TTS`
>
> - Focuses on architectural and training simplicity, similar to our goals.
> - Requires filler tokens and suffers from poor alignment learning, as shown in Figure 4 of the original paper.
> - We address alignment issues using context-aware batch expansion, which improves training stability and convergence without computational overhead.
> - Inference efficiency remains limited, with an RTF of 0.68 reported for E2-TTS in the F5-TTS paper—nearly 30× slower than SupertonicTTS, which achieves an RTF of 0.02.
>
> `About ZipVoice`
>
> - ZipVoice was not published in a peer-reviewed venue within two months prior to our submission.
> - According to the ICLR guidelines, authors are not required to compare their work to papers solely available on arXiv, so we are not obligated to include a comparison at this stage.
>
> `Speaker similarity`
>
> We respectfully clarify the following:
>
> - Our human preference tests show that:
>     - SupertonicTTS outperforms VoiceBox in speaker similarity.
>     - It also outperforms VALL-E in naturalness.
>     - It is competitive with CLaM-TTS and DiTTo-TTS across both perceptual dimensions.
> - Importantly, perceptual similarity is not fully captured by speaker verification scores. These scores depend on speaker embedding models that are often sensitive to synthesis artifacts and may not reflect human perception.
> - To illustrate the limitations of automatic verification, we conducted an additional analysis:
>     - We reconstructed identical utterances from the same speaker using three different vocoders, based on the test-clean subset of LibriSpeech.
>     - Speaker similarity between the original and reconstructed utterances was evaluated using WavLM-TDNN.
>     - Despite no change in speaker identity or content, the speaker similarity scores varied notably:
>
>
>         | Model | Speaker Similarity |
>         | --- | --- |
>         | BigVGAN-V2 | 0.9703 |
>         | HiFi-GAN | 0.9017 |
>         | Ours | 0.9318 |
>     - Human listeners judged both reconstructions as perceptually equivalent, yet the verification model produced a significant discrepancy in scores.
> - Based on this, we place greater emphasis on subjective preference evaluations, which we believe provide a more reliable and perceptually grounded assessment of speaker similarity.
> - We respectfully ask the reviewer to listen to our demo samples, which support these claims.
>
> `It is not clear that “training quickly with a very small model” is a central research problem for TTS`
>
> - Our contribution lies not in ‘training quickly,’ but in delivering strong performance with a small number of parameters and fast inference TTS within a streamlined pipeline.
> - This is particularly important, as it enables deployment on mobile devices, edge devices, and web browsers.
>
> `About streaming speech generation from an LLM`
>
> - LLMs are increasingly adopting parallel generation techniques based on diffusion frameworks, which aligns well with our approach.
> - Our model supports ultra-fast inference while maintaining a small GPU memory footprint, enabling a simpler pipeline alongside LLMs. Since LLMs already require substantial GPU memory, adding TTS often increases the load—but our model effectively minimizes this overhead.
>
> [1] Ji, Shengpeng, et al. "MobileSpeech: A Fast and High-Fidelity Framework for Mobile Zero-Shot Text-to-Speech." *Proceedings of the 62nd Annual Meeting of the Association for Computational Linguistics (Volume 1: Long Papers)*. 2024.

---

> > ### Comment · Reviewer_GKNu · 2025-11-28
> >
> > Thank you to the authors for the detailed rebuttal and for clarifying the motivations and design choices behind SupertonicTTS.
> >
> > * I agree that, according to the conference’s double-blind and archival policies, you are not strictly required to compare against ZipVoice, and I do not insist on adding that baseline for the purposes of policy compliance. However, simply improving RTF by reducing model size, without introducing genuinely new techniques for fast generation, does not constitute a strong novelty claim in my assessment. I also continue to have concerns about the data scalability and language extensibility of the proposed approach, which are not fully addressed by the current experiments.
> >
> > * Regarding speaker similarity metrics, in recent zero-shot TTS work it has effectively become standard practice to report objective speaker similarity scores on common benchmarks, typically using evaluation setups based on VALL-E, E2-TTS (LibriSpeech-PC), and Seed-TTS (Seed-TTS En/Zh eval). In my view, relying only on MOS–style subjective evaluations, without also providing objective metrics that are known to correlate reasonably well with human perception, is not sufficient. Otherwise, future papers may increasingly adopt heterogeneous, purely subjective protocols, making fair comparison across works more difficult and introducing additional noise due to evaluation design choices. Concretely, I encourage the authors to report SIM-O measured with a WavLM-TDNN setup on LibriSpeech and LibriSpeech-PC, in addition to the WER results already provided.
> >
> > * My reference to E2-TTS was to point out that the idea of performing zero-shot TTS directly from raw text input has already been publicly explored since June of last year. Furthermore, the final E2-TTS results reported by the authors of that work show improved alignment and pronunciation quality (e.g., WER 1.9 and SIM-O 0.67), so I do not think it is accurate to characterize E2-TTS as exhibiting “poor alignment learning” in its final form. Comparisons should be made against the official numbers reported in the original E2-TTS paper, rather than only against the re-implementation results in F5-TTS.
> >
> > * When I previously commented that the baselines feel insufficient, my concern was that the set of comparison models in the current paper does not include some of the more recent and competitive TTS systems. As a result, apart from sampling speed, I do not find the current results strong enough to conclude that the proposed model is clearly competitive in terms of TTS naturalness or overall quality at the current state of the field. In particular, several recent works seem relevant for comparison: MaskGCT (ICLR 2025), CosyVoice 2 (arXiv 2024.12, with 200+ citations), MELLE (ACL 2025), DiTAR (arXiv 2025.02), Spark-TTS (arXiv 2025.03), ...
> >
> > For these reasons, my overall assessment of the paper remains unchanged, and I maintain my original score.

---

> > > ### Author Response · Authors · 2025-12-03
> > >
> > > We sincerely thank the reviewer for the constructive follow-up.
> > >
> > >
> > > `simply improving RTF by reducing model size`
> > >
> > > SupertonicTTS is built on a new architecture across all major components (speech autoencoder, text-to-latent, duration predictor). Each module is newly designed to streamline the entire pipeline while delivering competitive performance. The improvements are not merely the result of reducing the parameter count. In combination with context-sharing batch expansion, latent compression, and new ConvNeXt-based building blocks, our model achieves strong performance with substantially faster inference.
> > >
> > >
> > > `SIM-O measured with a WavLM-TDNN`
> > >
> > > As noted in our earlier response, the conventional SIM-O metric using WavLM-TDNN does not yield reliable conclusions, as the measured similarity between the original and reconstructed audio varies even with the choice of vocoder. For example, the DiTTo-TTS paper reports that  VoiceBox achieves state-of-the-art SIM-O performance, yet its perceptual speaker similarity is clearly lower than that of CLaM-TTS and DiTTo-TTS when listening to the samples on their demo page. Our own preference test (Table 6) shows the same discrepancy. We therefore believe that WavLM-TDNN–based SIM-O can be misleading when evaluating speaker similarity in TTS systems, and we chose not to include it in the paper for this reason. While we agree on the importance of objective metrics, SIM-O with WavLM-TDNN appears insufficiently trustworthy at present.
> > >
> > >
> > > `About E2-TTS`
> > >
> > > We do not claim that our primary contribution is zero-shot TTS directly from raw text; rather, it is one of several outcomes enabled by our architectural innovations. DiTTo-TTS has already demonstrated zero-shot TTS from raw text without G2P by leveraging pretrained text encoders. Our architecture differs from E2-TTS and is designed to be more efficient and flexible, along with improved training techniques. For example, E2-TTS relies on filler tokens under the assumption that speech features outnumber text features. In contrast, SupertonicTTS compresses speech features along the temporal axis and can naturally handle scenarios where the opposite is true. This flexibility, combined with latent compression, leads to significantly faster inference than E2-TTS, which we believe the difference is far from minimal.

---

### Meta-Review · Area_Chair_UKyX · 2026-01-09

**Summary:**

This submission proposes SupertonicTTS, a streamlined and deployment-oriented zero-shot TTS pipeline consisting of (1) a speech autoencoder producing continuous latents, (2) a flow-matching text-to-latent model, and (3) an utterance-level duration predictor. The paper emphasizes efficiency through low-dimensional latents, temporal compression, and ConvNeXt blocks, while simplifying preprocessing by using character-level text and learning alignment via cross-attention. It also introduces context-sharing batch expansion as a training strategy to stabilize alignment and accelerate convergence with minimal memory/I/O overhead.
Overall, reviewers acknowledge the system’s practical appeal (small footprint and fast inference), but the discussion converges on two main blockers: unclear novelty/positioning in a crowded efficient zero-shot TTS landscape, and insufficiently standardized/complete evaluation to support claims of contemporary quality competitiveness beyond speed/size.

**Reviewer Concerns:**

Concerns clarified / partially addressed in the rebuttal
Streaming clarification: Authors clarify the decoder is causal and can support streaming audio output, while also noting first-byte benefits are limited because most compute lies in the text-to-latent stage; streaming text input is not considered.
RTF fairness / protocol sensitivity: Authors add an A100 measurement under a matched “10-second generation” setting to make RTF comparisons more interpretable.
Why not emphasize WavLM-TDNN speaker similarity: Authors provide evidence that speaker-similarity scores can vary notably across vocoders even on same-speaker reconstructions, motivating their preference for subjective evaluations.
Outstanding concerns (core issues remain)

Novelty/positioning remains weak: Multiple reviewers remain unconvinced that the paper introduces sufficiently novel techniques beyond combining known components; the paper’s differentiation vs recent efficient/non-AR/flow/diffusion TTS is not sharply articulated.
Evaluation gaps (major):
Missing standard objective speaker similarity reporting (e.g., SIM-O with WavLM-TDNN on common benchmarks such as LibriSpeech / LibriSpeech-PC), which reviewers consider standard for zero-shot TTS comparability across papers.
Baselines perceived as incomplete/outdated; reviewers explicitly request comparisons to more recent competitive systems (examples listed in discussion include MaskGCT, CosyVoice 2, MELLE, DiTAR, Spark-TTS).
Robustness evidence for “G2P-free” claims is not sufficiently stress-tested (e.g., hard sentences/edge cases).
Missing key ablations / empirical checks:
No direct ablation for Fourier head removal vs standard Vocos head under matched settings (quality/speed/memory trade-off).
Reviewers ask for an empirical validation of temporal compression “perfect inversion” .
Some protocol/conditioning/masking details are still considered under-specified for full reproducibility and evaluation interpretability.
Generalization evidence is limited: multilingual capability is mentioned but not demonstrated with systematic evaluation.

**Reviewer Scores:**

Reviewer GKNu: 2 (Reject), Confidence 5, and explicitly maintains the reject stance after discussion.
Reviewer ULR9: 4 (Marginally below threshold), Confidence 4; concerns focus on limited novelty and missing theoretical/ablation support for temporally compressed latents and generalization questions.
Reviewer vtnZ: 4 (Marginally below threshold), Confidence 4; acknowledges efficiency but emphasizes missing ablations (Fourier head, inversion check) and positioning clarity.
Reviewer 8Hfz: 6 (Marginally above threshold), Confidence 5, but indicates they would not mind rejection; also stresses missing ablations and broader comparisons.
Expected changes after rebuttal/discussion: No substantial upward changes are supported by the discussion record; one reviewer explicitly maintains reject, and the principal gaps (standard metrics, baselines, key ablations) remain.

---

### Decision · Program_Chairs · 2026-01-26

Reject